# Perinatal Exposure to Diesel Exhaust-Origin Secondary Organic Aerosol Induces Autism-Like Behavior in Rats

**DOI:** 10.3390/ijms22020538

**Published:** 2021-01-07

**Authors:** Tin-Tin Win-Shwe, Chaw Kyi-Tha-Thu, Yuji Fujitani, Shinji Tsukahara, Seishiro Hirano

**Affiliations:** 1Center for Health and Environmental Risk Research, National Institute for Environmental Studies, Tsukuba 305-8506, Japan; fujitani.yuji@nies.go.jp (Y.F.); seishiro@nies.go.jp (S.H.); 2Division of Life Science, Graduate School of Science and Engineering, Saitama University, Saitama City 338-8570, Japan; chawchaw25@nms.ac.jp (C.K.-T.-T.); stsuka@mail.saitama-u.ac.jp (S.T.)

**Keywords:** diesel exhaust, secondary organic aerosol, perinatal exposure, autism-like behavior, molecular analyses, rat

## Abstract

Autism spectrum disorder (ASD) is a neurodevelopmental disorder characterized by impaired social communication, poor social interactions, and repetitive behaviors. We aimed to examine autism-like behaviors and related gene expressions in rats exposed to diesel exhaust (DE)-origin secondary organic aerosol (DE-SOA) perinatally. Sprague–Dawley pregnant rats were exposed to clean air (control), DE, and DE-SOA in the exposure chamber from gestational day 14 to postnatal day 21. Behavioral phenotypes of ASD were investigated in 10~13-week-old offspring using a three-chambered social behavior test, social dominance tube test, and marble burying test. Prefrontal cortex was collected to examine molecular analyses including neurological and immunological markers and glutamate concentration, using RT-PCR and ELISA methods. DE-SOA-exposed male and female rats showed poor sociability and social novelty preference, socially dominant behavior, and increased repetitive behavior. Serotonin receptor (5-HT(5B)) and brain-derived neurotrophic factor (BDNF) mRNAs were downregulated whereas interleukin 1 β (IL-β) and heme oxygenase 1 (HO-1) mRNAs were upregulated in the prefrontal cortex of male and female rats exposed to DE-SOA. Glutamate concentration was also increased significantly in DE-SOA-exposed male and female rats. Our results indicate that perinatal exposure to DE-SOA may induce autism-like behavior by modulating molecules such as neurological and immunological markers in rats.

## 1. Introduction

Exposure to air pollutants may trigger neurodevelopmental and neurodegenerative disorders like autism spectrum disorder (ASD) and Alzheimer’s disease [1,2,3,4]. Epidemiological and experimental studies have indicated that associations exist between exposure to air pollutants during the brain developmental period and occurrence of ASD. In human studies, it was reported that maternal exposure to air pollution particulate matter (PM_2.5_) during pregnancy was associated with a greater risk of ASD in children [5,6]. Furthermore, case-control study conducted in Southwestern Pennsylvania indicated that PM_2.5_ exposure throughout pregnancy and until two years of age was associated with increased risk of ASD [7]. In animal studies, the male mice exposed to ambient ultrafine particles during the early postnatal period that is equivalent to the human third trimester expressed features of ASD including learning and memory deficiency, repetitive impulsive-like behavior, neuroinflammation, microglial activation, ventriculomegaly, and excitatory/inhibitory imbalance [8,9].

ASD is a neurodevelopmental disorder and is characterized by impaired social interaction, difficulties in language or communication, and repetitive behaviors. According to the Centers for Disease Control and Prevention (CDC), USA, recent ASD prevalence in the US is 1 in 59 children and it represents a public health issue and a large burden for education, social service, and economy [10]. Both genetic and environmental factors are considerable for ASD [11]. Recently, many researchers have paid attention to the association of exposure to air pollutants with an elevated risk of ASD. However, the precise etiology and pathophysiology of autism remains unclear. In this study, using diesel exhaust (DE)-origin secondary organic aerosol (SOA) as a model of air pollutants, we aimed to examine the effects of early life exposure to DE-SOA on autism-like behavior and neuroimmune responses including neurological and immunological biomarkers in rats.

PM_2.5_ consists of diesel exhaust particles which are major precursors of SOA formation. Previously, we have shown the effects of nanoparticle-rich diesel exhaust exposure on brain functions and behavior in adult mice [12,13,14,15,16,17]. Oxidation of volatile organic compounds in the atmosphere caused SOA formation [18]. Household and office appliances such as laser printers and copiers can also produce SOA [19,20]. However, there are limited reports for the effects of exposure to SOA on neurodevelopmental changes. This prompted us to study the effects of inhalation exposure to DE-SOA during the brain developmental period on ASD-like behavior.

We generated diesel exhaust-origin secondary organic aerosol (DE-SOA) by adding ozone to diesel exhaust and established the SOA inhalation chambers in our research institute. We studied the effect of early exposure to the air pollutant DE-SOA on neurotoxicity later in life using animal models. We showed that novel object recognition ability was impaired and accompanied by abnormally increased N-methyl-D-aspartate (NMDA) receptor expression in the hippocampus in adult male mice exposed to DE-SOA for three months [21]. Furthermore, one-month DE-SOA exposure in female mice caused a tendency to decrease maternal performance accompanied by decreased expression levels of estrogen receptor (ER)-α, and oxytocin receptor in [21]. In addition, a neonatal mouse model was established in our laboratory to examine early detection of learning disability by DE-SOA and found that olfactory-based spatial learning activity was impaired in neonatal mice [22]. Furthermore, we showed that exposure to DE-SOA during the brain developmental period impaired social behavior and altered social behavior-related gene expression in the hypothalamus of adult male mice [23]. Recently, we developed valproic acid (VPA)-induced autism model rats and showed the reduction of gamma amino butyric acid (GABA) synthetic enzyme glutamic acid decarboxylase (GAD) 67 protein in both male and female VPA-exposed rats. Thus, we suggest that the imbalance of glutamate/GABA was the possible mechanism for ASD [24]. Neuroligins are well-known cell adhesion molecules and the synaptic neuroligin level is strictly regulated by NMDA receptor functions in both wild type and Fmr1 knockout mice [25], suggesting that *Nlgn* and *Fmr1* are the genes that play a role in cognitive functions. Moreover, recent study has indicated that *Fmr1*, *Cyfip1*, and *Nlgn3* are expressed at the synapse and play a role in structural and physiological plasticity and involved in ASD. [26]. Although we showed that developmental period exposure to DE-SOA induced impaired social behavior in mouse models [23], we need to compare autism-like behaviors in VPA-induced autism model rats [24]. Thus, we examined whether the developmental period exposure to DE-SOA induced autism-like behavior in rat models in the present study. 

Some animal studies have shown that inhaled 13C particles from graphite rods in rats and intranasally instilled manganese oxide particles in monkeys translocate to the central nervous system via the olfactory nerve [27,28]. In addition, our previous data showed that increased proinflammatory cytokine mRNA expressions were observed in the olfactory bulb of mice intranasally instilled with carbon black nanoparticles [29]. Therefore, we hypothesized that the toxic constituents from DE-SOA may translocate to the brain via the olfactory nerve pathway or systemic circulation and induce neuroinflammation and affect social behavior. The purpose of this study was to detect the effects of perinatal (gestational and neonatal) exposure to DE-SOA on autism-like behavior and neuroimmune response in rats.

## 2. Results

### 2.1. General Toxicity Assessment

To determine the general toxicity of DE or DE-SOA, body and brain weights of the rats were measured at the time of sampling. Although body weight and brain weight of DE-exposed male and female rats were increased compare to the control and DE-SOA exposed groups, no statistically significant difference was observed among the groups (Table 1).

### 2.2. DE-SOA Exposure Impaired Sociability and Social Novelty Preference 

Sociability test: The control rats approached the stranger 1 (S1) cup more times than the empty (E) cup (*p* < 0.05). DE-exposed rats equally approached the E and S1 cups. DE-SOA-exposed rats approached the E cup more times than the S1 cup (*p* < 0.05). Both male and female rats showed a similar pattern (Figure 1A).

Social novelty preference test: The control rats approached the stranger 2 (S2) cup more times than the S1 cup (*p* < 0.05). DE-exposed rats equally approached the S1 and S2 cups. DE-SOA-exposed rats approached the S1 cup more times than the S2 cup (*p* < 0.05). Both male and female rats showed a similar pattern (Figure 1B).

It was suggested that the control rats recognized novel or new ones and the DE- or DE-SOA-exposed rats had a poor ability to discriminate between familial and novel ones.

### 2.3. DE-SOA Exposure Induced Social Dominance Behavior

In social dominance tube test, male rats exposed to DE-SOA showed increased win% compared to the control and DE-exposed rats (*p* < 0.01, Figure 2A) and female rats exposed to DE-SOA showed increased win% compared to the control rats (*p* < 0.01, Figure 2B).

It was suggested that male and female rats exposed to DE-SOA had social dominance or aggressive behavior.

### 2.4. DE-SOA Exposure Induced Repetitive Behavior

In the marble burying test, male rats exposed to DE or DE-SOA showed significantly increased percentage of buried marble compare to the control rats (*p* < 0.05, Figure 3A) and female rats exposed to DE-SOA showed significantly increased percentage of buried marble compare to the control rats (*p* < 0.01, Figure 3B).

These results indicate that male and female rats exposed to DE-SOA performed repetitive behavior.

### 2.5. DE-SOA Exposure Altered the mRNA Expressions of Social Behavior-Related Genes and Neurotrophic Factor in the Prefrontal Cortex

We investigated the effect of DE or DE-SOA exposure on social behavior-related genes and neurotrophic factor mRNA expressions in the prefrontal cortex of rats. We found that the expression levels of 5-hydroxytryptamine (serotonin) receptor 5B (5-HT5B) were significantly reduced in male and female rats exposed to DE-SOA compared with the corresponding control (*p* < 0.05, Figure 4A,B). *Nlgn3* is a gene associated with ASD. We found *Nlgn3* mRNA expression was significantly decreased in male rats exposed to DE-SOA, but not in the female rats (*p* < 0.05, Figure 4C,D). We also observed that the expression levels of brain-derived neurotrophic factor (BDNF) were significantly lower in male and female rats exposed to DE-SOA compared with the corresponding control (*p* < 0.05, Figure 4E,F).

### 2.6. DE-SOA Exposure Altered the mRNA Expressions of Inflammatory Markers in the Prefrontal Cortex

To detect the DE- or DE-SOA-induced inflammation in the prefrontal cortex, we also investigated the inflammatory markers such as interleukin (IL)-1β, tumor necrosis factor (TNF) α, and cyclooxygenase (COX)2 in the prefrontal cortex of rats. We found that the expression level of IL-1β mRNA was remarkably increased in male and female rats exposed to DE-SOA compared with the corresponding control (*p* < 0.01; *p* < 0.05, Figure 5A,B). No significant changes of TNF-α and COX2 were found. We also observed that the expression levels of heme oxygenase (HO)-1 were significantly higher in male and female rats exposed to DE-SOA compared with the corresponding control (*p* < 0.01, *p* < 0.05, Figure 5C,D).

### 2.7. DE-SOA Exposure Increased the Level of Neuronal Excitatory Marker in the Prefrontal Cortex 

Excessive glutamate level is the indicator of neurotoxicity. We measured glutamate concentration in the prefrontal cortex and found that glutamate secretion was increased remarkably in both male and female rats exposed to DE-SOA compared with the control rats (* *p* < 0.05) (Figure 6).

## 3. Discussion

In our previous study, we showed that developmental exposure to DE-SOA impaired sociability and social novelty preference and the hypothalamic expression of social behavior-related genes estrogen receptor alpha and oxytocin receptor in male mice [23]. Recently, we used VPA-induced autistic rat models to examine the role of excitatory and inhibitory neurotransmitters, especially GABA synthetic enzyme in the mechanism of ASD [24]. To compare the autism-like behavior in rats induced by VPA [24] with perinatal exposure to DE-SOA, we used rat models in the present study to examine the ASD related social behaviors and related gene expressions in the ASD-related brain region, prefrontal cortex. In the three-chamber social behavioral test, we found same pattern of sociability and social novelty preference were observed in male mice and rats. Our targets in the previous mice study and the present rat study were different in animal strain, target brain area, and target genes. The other reasons for using rat models in the present study were to examine the ASD-related social behaviors other than by three-chamber test, the sexually different effects, and the role of social behavior-related genes such as serotonin, BDNF, and inflammatory marker IL-1β.

The major findings of the present study were that the perinatal exposure to DE-SOA induces (1) impairment of sociability and social novelty preference, increased social dominance behavior, and increased repetitive behavior; (2) downregulation of social behavior-related gene serotonin receptor 5HT5B, *Nlgn3*, and neurotrophic factor BDNF; (3) upregulation of proinflammatory cytokine IL-1β and oxidative stress marker HO-1; and (4) increased neurotoxic substance glutamate concentration in male and female rats. Our findings suggest that the perinatal exposure to DE-SOA induces autism-like behavior in rats by triggering neuroinflammation and neurodevelopmental disorder via neurological and immunological biomarkers in the brain.

ASD is a neurodevelopmental disorder and is characterized by impaired social interaction, difficulties in language or communication, and repetitive behaviors. In the present study, impairment of social interaction, increased social dominance, and increased repetitive behavior were observed in DE-SOA-exposed male and female rats. The exact causes of ASD are not known, and both genetic and environmental factors are suggested to contribute to ASD. Fragile X syndrome, Rett syndrome, or cytogenetic abnormalities are reported to be associated with ASD [30,31]. *NLGN3*, *NLGN4X*, and *SHANK3* are synaptic genes, and the mutation of these genes is involved in idiopathic autism [32,33,34,35]. Regarding environmental factors, maternal infection in the first trimester [36] and poor pregnancy outcome such as low birth weight, preterm birth, small-for-age, maternal hemorrhage, socioeconomic status, drugs, and environmental toxic substance exposure are risk factors for ASD [37,38]. Maternal education level with poor socioeconomic status may influence the risk for ASD. Exposure to valproate, an epileptic drug, during first trimester of pregnancy is a risk factor for ASD. It was reported that gestational exposure to valproate causes an 8-fold-increased risk to have ASD in children [39,40]. Previously, we demonstrated that developmental exposure to valproic acid induced poor social behavior in adult rats by modulating the expression levels of social behavior-related genes and inflammatory mediators accompanied with changes in GABA enzyme glutamic acid decarboxylase 67 (GAD67) in the hippocampus [24]. In that study we showed that GAD67 deficiency may be associated with glutamate/GABA imbalance in the autistic brain. We have also reported that early life exposure of BALB/c mice to DE-SOA may affect their late-onset hypothalamic expression of social behavior-related genes (ERα and oxytocin receptor) and induced impaired social behavior [23]. Taken together, environmental factors such as maternal infection, nutrition, antenatal care, education, drug usage, and exposure to chemicals or pollutants during pregnancy are risk factors for ASD.

Recently we reported that developmental exposure to VPA and arsenic impaired social behavior by modulating serotonin receptors and decreasing BDNF [24,41]. In the present study, we found downregulation of mRNA expression levels of social behavior-related gene serotonin receptor 5HT5B and neurotrophic factor BDNF in the prefrontal cortex of male and female rats exposed to DE-SOA. Among serotonin receptors, 5HT5B was selected because this receptor is involved in social behavior [42]. BDNF is involved in synaptic plasticity which is critical for learning and memory functions. Downregulation of BDNF may impair synaptic plasticity and may also affect development and function of serotonin neurons [43].

Immune dysfunction such as abnormalities in T cells, B cells, and NK cells; the production of autoantibodies; and increasing proinflammatory cytokines were reported in autism [44,45,46]. Patients with ASD showed an increase of proinflammatory and regulatory cytokines in the cerebrospinal fluid [47]. In the present study, proinflammatory cytokine IL-1β and oxidative stress marker HO-1 mRNAs were increased significantly in the prefrontal cortex of male and female rats exposed to DE-SOA. Regarding brain immune cells, it has been reported that microglial activation was observed in the brain, especially in the prefrontal cortex, cerebellum, and cerebral white matter of patients with ASD [48,49]. These results are consistent with our present findings showing activation of microglia in the prefrontal cortex of male rats exposed to DE-SOA. In addition, we found that mast cell expression was increased in the prefrontal cortex of male rats exposed to DE-SOA. We found sex-specific effects of DE-SOA on microglia and mast cell activation.

Glutamate is an excitatory amino acid neurotransmitter and excessive glutamate release caused excitotoxicity and induced neuronal death [50]. We found that glutamate concentration was increased significantly in the prefrontal cortex of male and female rats exposed to DE-SOA perinatally. Increased glutamate levels in DE-SOA-exposed rats may cause blockage of re-uptake by glutamate transporters in the presynaptic neurons, and decreased GABA synthetic enzymes are possible causes of increased glutamate concentration. Although we did not examine GABA, an inhibitory amino acid neurotransmitter, or its synthetic enzyme GAD level, excitatory-inhibitory imbalance may trigger autism-like behavior in DE-SOA-exposed rats.

Human studies have indicated that pregnant mothers who live or work near highways and busy roads were prone to exposure to air pollutants including diesel exhaust particles and increased the risk of ASD [51,52,53,54]. Some studies have indicated that the critical window for prevalence of childhood ASD was the third trimester of pregnancy [9,55,56] because cortical synaptogenesis reaches peak at that period [5]. In addition, infection-induced maternal immune activation triggers neuroinflammation in the placenta and fetal brain which causes ASD-like behavior in childhood [57]. Taken together, location of residential area, exposure dose, gestational period of exposure, maternal risk factors, and socioeconomic status of parents are predisposing factors for air pollution-induced increased risk of ASD.

We found no effect of DE on social novelty preference performance and social behavior-related genes such as 5HT5B, BDNF, IL-1β, HO1, and glutamate in the present study. The difference between DE and DE-SOA was constituents, and there were 1000 constituents from DE and DE-SOA; we could not say exactly which constituents influenced the behavior and related biomarkers. Recently, we have been investigating the candidate chemicals which have adverse effects in DE and DE-SOA in inhalation chambers. 

Sex-specific effects were observed in ASD, and boys have 5 times greater risk for ASD than girls. In our previous study, impairment of social behavior and related gene expression changes were prominent in male rats of the VPA-induced autism model [24]. A male-specific upregulation of transcription factors such as *Pax6, Tbr1, Tbr2, Sp1*, and *Creb1* was found in postnatal day (PND) 3 mice exposed to DE from embryonic day 0 to PND 3 [58]. In the present study, no remarkable statistical differences of ASD-related behaviors were observed between male and female rats. However, in the three-chamber social behavioral test, during sociability performance, the total exploration times were 210 s (control), 110 s (DE), and 125 s (DE-SOA) in male rats and 180 s (control), 120 s (DE), and 113 s (DE-SOA) in female rats. The pattern was similar but less pronounced differences were observed in female rats. During social novelty preference performance, DE- and DE-SOA-exposed male and female rats did not prefer stranger 2 rats and that novelty preference was lower in male rats compared to female rats. Regarding ASD-related genes, we found *Nlgn3* mRNA expression was significantly decreased in DE-SOA-exposed male rats, but not in the female rats in the present study. A recent report has indicated that developmental exposure to PM_2.5_ caused autism-like behaviors, such as poor social interaction and repetitive behavior, accompanied by lower oxytocin receptor (OXTR) protein level, catalase activity, and glutathione (GSH) concentration, in 7-week-old male rats [59]. Different hormonal milieu, receptor type and function, and maturation of neural network may contribute the sex-specific effects in ASD. Moreover, compensatory effects of deficient ASD-related genes may protect females from ASD [60]. Finally, our results indicate that perinatal exposure to DE-SOA may induce autism-like behavior by modulating molecular levels such as neurological and immunological markers in rats.

## 4. Materials and Methods 

### 4.1. Animals

Forty-eight timed pregnant Sprague–Dawley rats (gestational day (GD) 8) were purchased from Oriental Yeast Co., Ltd. (Tokyo, Japan) and exposed to clean air (control, *n* = 16), DE (*n* = 16), and DE-SOA (*n* = 16) from GD 14 to postnatal day (PND) 21 in whole-body exposure chambers. Food and water were given ad libitum. Date of birth was recorded as PND 0 and the offspring were housed in cages with dam under controlled environmental conditions (temperature, 22 ± 0.5 °C; humidity, 50 ± 5%; lights on 07:00–19:00 h). The pregnant rats were exposed in the exposure chamber for 5 h per day (from 10:00 p.m. to 3:00 a.m.), 5 days a week excluding weekends from gestational day 14 to postnatal day 21 with their pups. 

The numbers of pups born were 128 (60 male and 68 female) in the control, 130 (68 male and 62 female) in the DE group, and 135 (63 male and 72 female) in the DE-SOA group. We used 3 male and 3 female pups from each dam (total 48 male and 48 female pups). Among them, 16 male and 16 female rats were used for the three-chamber social behavior test and marble burying test, 14 were used for neurochemical analyses, and 2 were used for immunohistochemical analyses. Another 16 male and 16 female rats were used as test animals for the social dominance test and another 16 male and 16 female rats were used for pair matching. Furthermore, 4 male and 4 female rats from each group were used as age-matched stranger 1 and stranger 2.

The pups were weaned at PND 21 and 3~4 pups of same sex were housed in a plastic cage. The experimental design is depicted in Figure 7. Social behavioral tests were performed at 10~13 weeks old. Behavioral testing was performed between 09:00 and 13:00 h. Before performing each test, the apparatus to be used was cleaned with 70% ethanol. After completing the social behavioral test, the rats were sacrificed under deep pentobarbital anesthesia and the left and right prefrontal cortex were collected from each group and frozen quickly in liquid nitrogen, then stored at −80 °C until the extraction of the total RNA. The experimental protocols were approved by the Ethics Committee of the Animal Care and Experimentation Council of the National Institute for Environmental Studies (NIES), Japan (AE-19-36, 19 December 2019; AE-20-05, 9 March 2020). All efforts were made to minimize the number of animals used and their suffering.

### 4.2. Preparation of Exposure Chambers for Clean Air, DE, or DE-SOA 

The whole-body inhalation exposure chambers for clean air, DE, or DE-SOA were generated at the National Institute for Environmental Studies, Japan, as described previously [21,61]. Briefly, an 81-diesel engine (J08C; Hino Motors Ltd., Hino, Japan) was used to generate diesel exhaust. The engine was operated under a steady-state condition for 5 h a day. Our driving condition of the diesel engine was not simulated to any special condition as in the real world. The engine operating condition (2000 rpm engine speed and 0 Nm engine torque) promoted the generation of high concentrations of nano-size particles. There were three chambers: a control chamber receiving clean air filtered through a high efficiency particulate air (HEPA) filter and a charcoal filter (referred to as “clean air”), the diluted exhaust (DE which was without mixing O_3_) chamber, and the DE-SOA chamber, which was generated by mixing DE with ozone at 0.6 ppm after secondary dilution. Secondary dilution ratio in the DE and DE-SOA chambers were the same, which resulted in the same particle and gaseous concentrations when O_3_ was not mixed. Actually, the concentrations of particles in DE-SOA were higher when O_3_ was mixed, and the concentrations of DE and DE-SOA were 101 ± 9 μg/m^3^ and 118 ± 23 μg/m^3^, respectively. The increased mass concentration was due to the generation of secondary particles. The temperature and relative humidity inside each chamber were adjusted to approximately 22 ± 0.5 °C and 50 ± 5%, respectively. The particle characteristics inside the exposure chamber are shown in Table 2. In detail, sample air was taken from the inhalation chamber (2.25 m^3^) using stainless steel tubing. The gas concentrations (CO, CO_2_, NO, NO_2_, and SO_2_) were monitored using a gas analyzer (Horiba, Kyoto, Japan). CO and NO_x_ concentrations in both chambers were similar, but NO and NO_2_ were different from each other because NO was oxidized to NO_2_ by reaction with O_3_. The particle size distributions were measured using a scanning mobility particle sizer (SMPS 3034, TSI Instruments, Shoreview, MN, USA). The modal sizes of the particles used in the present study were 22.69 ± 1.47 nm for DE and 24.45 ± 1.21 nm for DE-SOA. The particles were collected using a Teflon filter (FP-500; Sumitomo Electric, Osaka, Japan) and a Quartz fiber filter (2500 QAT-UP; Pall, Pine Bush, NY, USA), and the particle mass concentrations were measured using a Teflon filter. The particle weights were measured using an electrical microbalance (UMX 2, Mettler-Toledo, Columbus; OH, USA; readability 0.1 μg) in an air-conditioned chamber (CHAM-1000; Horiba) under constant temperature and relative humidity conditions (21.5 °C, 35%). For the Quartz fiber filter, the quantities of elemental carbon (EC) and organic carbon (OC) were determined using a carbon analyzer (Desert Research Institute, Reno, NV, USA). EC to OC ratios in the present study were 0.15 ± 0.06 for the control chamber, 0.36 ± 0.03 for the DE chamber, and 0.38 ± 0.03 for the DE-SOA exposure chamber. 

### 4.3. Behavioral Assessment

Sixteen male and female rats were used for social behavior analyses.

#### 4.3.1. Sociability and Social Novelty Preference 

Sociability and preference for social novelty tests were performed as reported previously [21]. The apparatus used was a rectangular, three-chambered Plexiglas box (100 cm × 100 cm × 35 cm), with equal-sized chambers. The dividing partitions were also made of clear Plexiglas, with small doorways on each (10 cm × 10 cm) that allowed free access for the animals among the chambers. Wired cups (diameter, 15 cm; height, 30 cm) were placed in each of the side chambers to house unfamiliar animals. For habituation, the subject rats from three different groups (DE- or DE-SOA-exposed and the control rats) were placed in the middle chamber and allowed to explore for 5 min. During the habituation phase, the wired cup in each of the side chambers was empty (E). Following habituation, for the sociability test, an unfamiliar rat (stranger 1 (S1), age-matched rat) was placed in the wired cup in one of the side chambers; the subject rats were allowed to explore for 10 min. The location of stranger 1 in the left- or right-side chamber was systematically alternated between trials. The social novelty preference test was performed immediately after the sociability test. For this test, another unfamiliar rat (stranger 2 (S2), age-matched rat) was placed in the wired cup on the other side that had been empty during the first 10 min session, and the subject rat was allowed to explore the two strangers for 10 min. The time spent exploring the wired cups on either side was measured. The time that the subject rat spent exploring the wired cup was measured as the time spent with its head facing the cup from a distance of within 1 cm.

#### 4.3.2. Social Dominance Behavior (Tube Test)

Social dominance was tested in a transparent Plexiglas tube measuring 45 cm in length and 4 cm in (inside) diameter, a size just sufficient to permit one rat to pass through without reversing direction [62]. For training, each rat was released at alternating ends of the tube and allows to run through the tube. Each animal was given five training trials on each of two successive days. For the social dominance test, animals were placed at opposite ends of the tube and released simultaneously. An animal was declared the “winner” when its opponent backed out of the tube. The maximum test time was set to 2 min. The tube was cleaned with 70% ethanol before each trial.

#### 4.3.3. Marble Burying Test

The marble burying test is a useful model of anxiety-like behavior and repetitive behavior. Each rat was placed for 20 min into a clean rat cage (40 cm × 24 cm × 15 cm) with 5 cm deep bedding and 20 glass marbles placed in a regular pattern and evenly spaced. The number of marbles that were buried at least 2/3 of the area by the rat was measured. 

#### 4.3.4. Quantification of mRNA Expression Levels

After completion of the behavioral tests, 13-week-old male and female rats (*n* = 14 from each group) were sacrificed under deep pentobarbital anesthesia and the left prefrontal cortex was collected from each group for mRNA analyses. Briefly, the total RNA was extracted from the prefrontal cortex samples using the BioRobot EZ-1 and EZ-1 RNA tissue mini kits (Qiagen GmbH, Hilden, Germany). Then, the purity of the total RNA was examined, and the quantity was estimated using the ND-1000 NanoDrop RNA Assay protocol (NanoDrop, Wilmington, DE, USA), as described previously [13]. Next, we performed first-strand cDNA synthesis from the total RNA using SuperScript RNase H−Reverse Transcriptase II (Invitrogen, Carlsbad, CA, USA), according to the manufacturer’s protocol. We examined the mRNA expression levels using RT-PCR (Light Cycler 96, Roche, Germany). The tissue 18S rRNA level was used as an internal control. The primer sequences used in the present study are shown below. Some primers (5-HT5B, NM_024395; BDNF, NM_012513; IL-1β, NM_008361; COX2, NM_011198; HO-1, NM_010442; and *Nlgn3*, NM_134336) were purchased from Qiagen, Sample and Assay Technologies. Other primers were designed in our laboratory as follows: 18S (forward 5′-TACCACATCCAAAAGGCAG-3′, reverse 5′-TGCCCTCCAATGGATCCTC-3′) and TNF-α (forward 5′-GGTTCCTTTGTGGCACTTG-3′, reverse 5′-TTCTCTTGGTGACCGGGAG-3′). Data were analyzed using the comparative threshold cycle method. Then, the relative mRNA expression levels were expressed as mRNA signals per unit of 18S rRNA expression.

#### 4.3.5. Measurement of Glutamate Concentration 

After completing the social behavioral test, the rats (*n* = 14 from each group) were sacrificed under deep pentobarbital anesthesia and the right prefrontal cortex was collected from six male and female rats of each group and frozen quickly in liquid nitrogen, then stored at −80 °C until protein analysis. The right prefrontal cortexes from 6 rats of each male and female group were homogenized in a falcon tube containing 10 mL of cool sterile saline and centrifuged at 3000 rpm/min for 5 min at 4 °C. The supernatant was used for subsequent glutamate detection by glutamate research ELISA assay kit (Ref: BA E-2300, Neuroscience. Inc., Tokyo, Japan) according to the manufacturer’s instructions.

### 4.4. Statistical Analysis 

All the data were expressed as the mean ± standard error (SE). The statistical analysis was performed using the StatMate II statistical analysis system for Microsoft Excel, Version 5.0 (Nankodo Inc., Tokyo, Japan). The data were analyzed using a one-way analysis of variance with a post-hoc analysis using the Bonferroni/Dunn method. Differences were considered significant at *p* < 0.05.

## 5. Conclusions

Developmental exposure to DE-SOA induced autism-like behaviors such as poor social interaction, social dominance, and repetitive behavior in male and female rats. These poor social behaviors are accompanied by changes of molecular markers such as neurological and immunological biomarkers in the prefrontal cortex. It is possible that the toxic substances from DE-SOA may translocate to the fetal brain via the olfactory nerve route or systemic circulation and induce neuroinflammation. Neuroimmune interaction, synaptic dysfunction, immune dysregulation, and excitatory-inhibitory (E-I) imbalance are major contributing factors for the development of ASD in response to environmental pollutant exposure. Further studies are needed to elucidate the role of other social behavior-related neurological and immunological biomarker changes in environmental pollutant-induced ASD-like behavior.

## Figures and Tables

**Figure 1 ijms-22-00538-f001:**
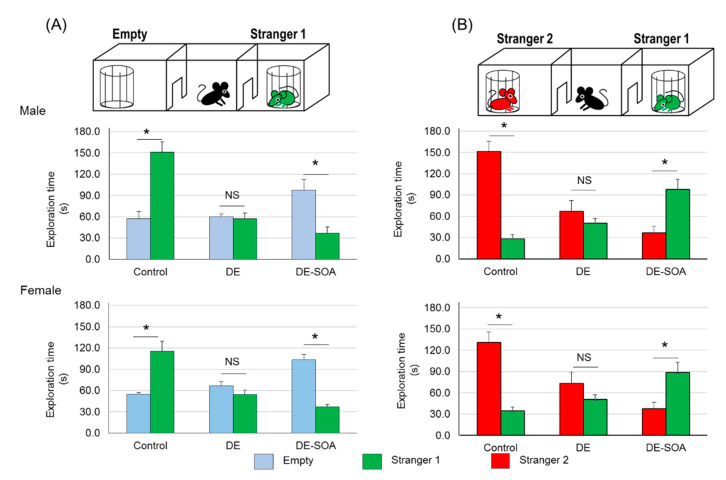
Assessment of social behavior using three-chamber social behavior test. (**A**) sociability test and (**B**) social novelty preference test in the control and in the diesel exhaust (DE)- and secondary organic aerosol (DE-SOA)-exposed male and female rats (*n* = 16, * *p* < 0.05). NS: not significant.

**Figure 2 ijms-22-00538-f002:**
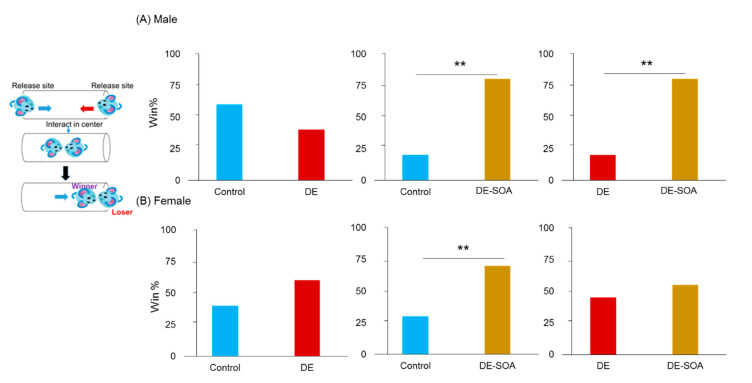
Assessment of social dominance behavior using tube test. Social dominance behavior in (**A**) male and (**B**) female rats of the control and the DE- and DE-SOA-exposed rats (*n* = 16, ** *p* < 0.01 vs. corresponding control or DE-exposed rats).

**Figure 3 ijms-22-00538-f003:**
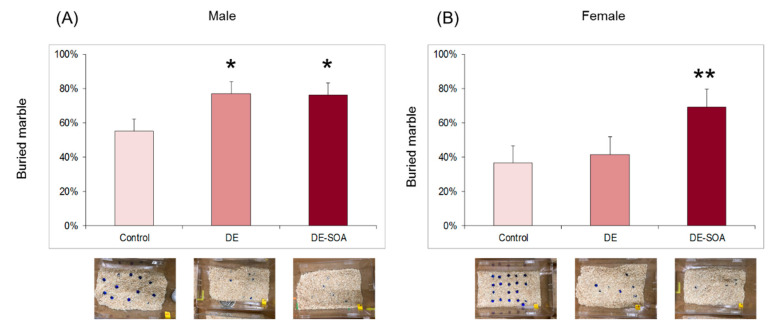
Assessment of repetitive behavior using marble burying test. Repetitive behavior in (**A**) male and (**B**) female rats of the control and the DE- and DE-SOA-exposed rats (*n* = 16, ** *p* < 0.01, * *p* < 0.05 vs. corresponding control).

**Figure 4 ijms-22-00538-f004:**
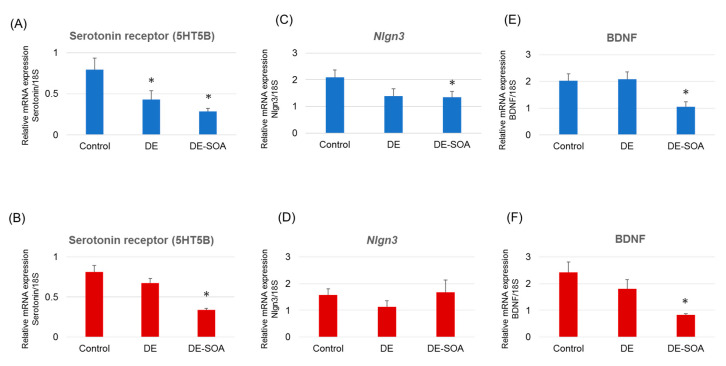
mRNA expression levels of neurological biomarkers. (**A**,**B**) Serotonin receptor 5HT5B, (**C**,**D**) *Nlgn3*, and (**E**,**F**) brain-derived neurotrophic factor (BDNF) in the prefrontal cortex of the control and the DE- and DE-SOA-exposed male and female rats. (*n* = 14, control; * *p* < 0.05 vs. control).

**Figure 5 ijms-22-00538-f005:**
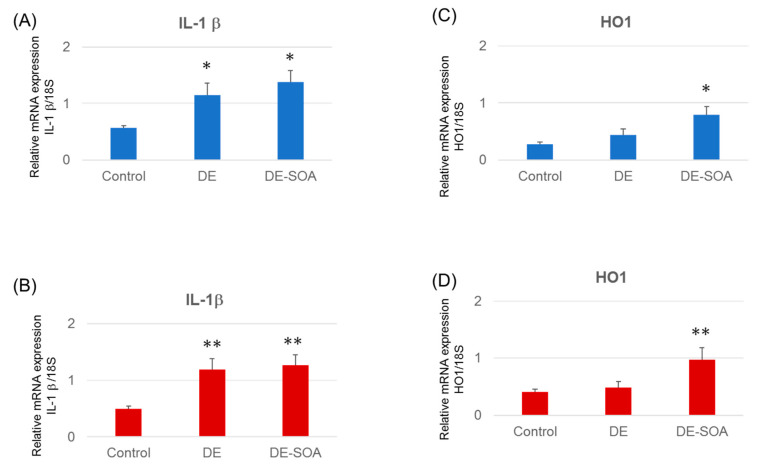
mRNA expression levels of immunological biomarkers. (**A**,**B**) IL1-β and (**C**,**D**) HO-1 in the prefrontal cortex of the control and the DE- and DE-SOA-exposed male and female rats. (*n* = 14, ** *p* < 0.01 vs. control; * *p* < 0.05 vs. control).

**Figure 6 ijms-22-00538-f006:**
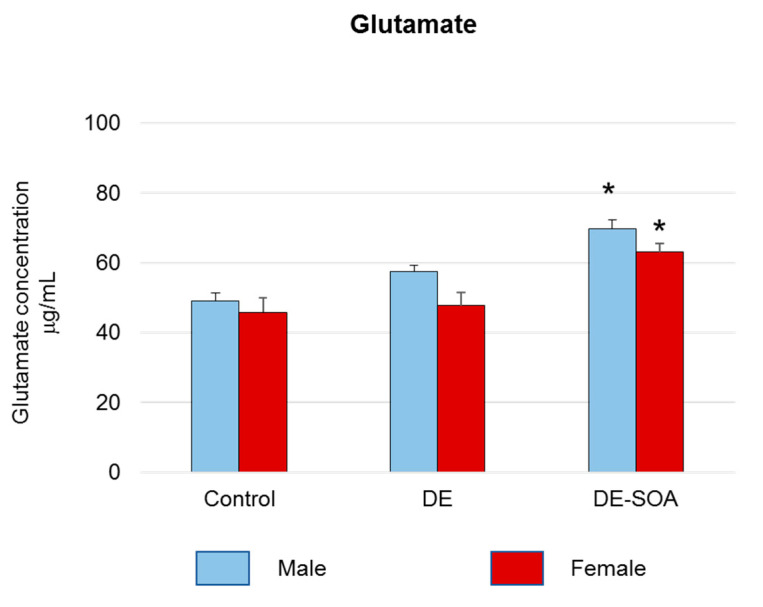
Assessment of neurotoxic substance in the brain. Glutamate concentration in the prefrontal cortex of the control and the DE- and DE-SOA-exposed male and female rats (*n* = 6, * *p* < 0.05 vs. corresponding control).

**Figure 7 ijms-22-00538-f007:**
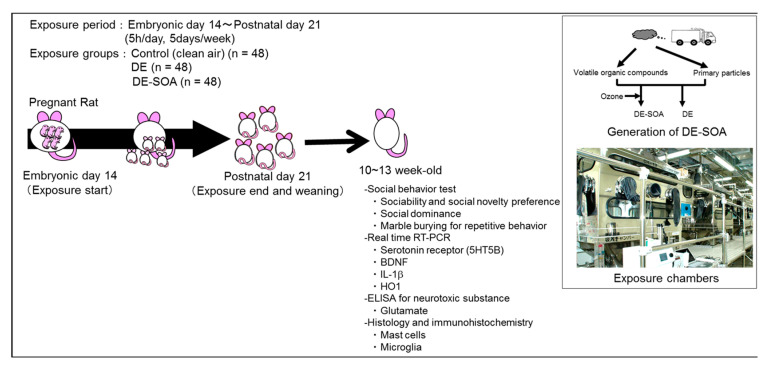
Experimental protocol including generation of DE-SOA and exposure chambers. Forty-eight timed pregnant Sprague–Dawley rats were exposed to clean air (control, *n* = 48), DE (*n* = 48), or DE-SOA (*n* = 48) from gestational day (GD) 14 to postnatal day (PND) 21 in the whole-body exposure chambers. Sixteen male and female offspring from the control, DE, and DE-SOA groups were used for social behavioral tests at 10~13 weeks old. After completing the social behavioral test, brain samples were collected for molecular analyses.

**Table 1 ijms-22-00538-t001:** Body weight and brain weight of male and female offspring rats.

	Male Rat		Female Rat	
	Body Weight (g)	Brain Weight (mg)	Body Weight (g)	Brain Weight (mg)
Control	298.0 ± 7.0	1917.4 ± 22.6	192.4 ± 9.0	1844.3 ± 27.0
DE	315.4 ± 3.7	2059.1 ±37.6	219.0 ± 7.7	1900.6 ± 51.5
DE-SOA	296.6 ± 5.4	1936.3 ± 31.5	203.4 ± 6.0	1762.0 ± 22.4

**Table 2 ijms-22-00538-t002:** Particles and gaseous compounds in the exposure chambers.

**Diesel Exhaust Particles**						**Temperature**	**Relative Humidity**
	**Size (nm)**	**Particle Number (cm^−3^)**	**Concentration (μg/m^3^)**	**EC/OC**	**WSOC/OC**	**(°C)**	**(%)**
Clean air	NA	2.36 × 10^2^ ± 1.31 × 10^1^	5.8 ± 8.00	0.00 ± 0.00	0.24 ± 0.10	23.15 ± 0.22	50.47 ± 1.20
DE-SOA	24.45 ± 1.21	2.50 × 10^6^ ± 7.51 × 10^4^	118.15 ± 22.52	0.00 ± 0.00	0.07 ± 0.01	23.12 ± 0.15	50.71 ± 1.48
DE	22.69 ± 1.47	2.25 × 10^6^ ± 8.61 × 10^4^	101.28 ± 8.73	0.00 ± 0.00	0.06 ± 0.01	22.74 ± 0.13	51.49 ± 2.05
**Gaseous Compounds**							
	**CO (ppm)**	**SO_2_ (ppm)**	**NO_x_ (ppm)**	**NO_2_ (ppm)**	**NO (ppm)**	**O_3_ (ppm)**	**CO_2_ (%)**
Clean air	0.15 ± 0.02	0.00 ± 0.00	0.00 ± 0.00	0.00 ± 0.00	0.00 ± 0.00	ND	0.06 ± 0.00
DE-SOA	2.79 ± 0.04	0.00 ± 0.00	1.48 ± 0.03	1.14 ± 0.02	0.34 ± 0.03	0.06 ± 0.00	0.08 ± 0.00
DE	2.94 ± 0.05	0.01 ± 0.00	1.64 ± 0.03	0.58 ± 0.01	1.05 ± 0.03	ND	0.08 ± 0.00

NA, not available; ND, not detectable.

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
