# Peer review of "Perinatal Exposure to Diesel Exhaust-Origin Secondary Organic Aerosol Induces Autism-Like Behavior in Rats"

_ijms, 2021, doi:10.3390/ijms22020538_

Round 1
Reviewer 1 Report
The research article entitled “Perinatal exposure to diesel exhaust origin secondary organic aerosol induces autism-like behavior in rats” by Tin-Tin Win-Shwe and colleagues is an interesting article to read. The authors did experiments on mice to show the deleterious effect of automobile exhaust on social behaviour. Authors observed diesel exhaust and origin secondary organic aerosol induced autism-like behavior. The article is fascinating, well written, analyzed and discussed properly. In my view the article can be accepted for publication with minor corrections. Some of the point may improve the quality of MS 1. Nlgn3 expression has been discussed in result and discussion section; however one or two line about Nlgn3 should be discussed in introduction section. 2. Line no 175-177 “These studies prompted…” can be rewritten for clarity of message. 3. Line no 151-152 “We did not know… not clear, please rewrite. 4. Line no 77-80 and 198-199, authors wrote about unpublished data.. These days many journals do not allow to include unpublished data..I my view authors should remove findings of unpublished data. 5. Recent reports are few….it is good to added/cite recent findings.Author Response
Responses to the Reviewers’ comments
Manuscript ID: ijms-1056223
Reviewer # 1
The research article entitled “Perinatal exposure to diesel exhaust origin secondary organic aerosol induces autism-like behavior in rats” by Tin-Tin Win-Shwe and colleagues is an interesting article to read. The authors did experiments on mice to show the deleterious effect of automobile exhaust on social behaviour. Authors observed diesel exhaust and origin secondary organic aerosol induced autism-like behavior. The article is fascinating, well written, analyzed and discussed properly. In my view the article can be accepted for publication with minor corrections. Some of the point may improve the quality of MS.
- Nlgn3 expression has been discussed in result and discussion section; however one or two line about Nlgn3 should be discussed in introduction section.
As commented by the Reviewer, we added about Nlgn3 in INTRODUCTION session page 2, line 77-81 in our revised manuscript as follows;
Neuroligins are well known cell adhesion molecules and synaptic neuroligin level is strictly regulated by NMDA receptor functions in both wild type and Fmr1 knockout mice Chmielewska et al., 2019) suggesting that these two genes play a role in cognitive functions. Moreover, recent study has indicated that Fmr1, Cyfip1 and Nlgn3 are expressed at synapse and triggering structural and physiological plasticity and seem to contribute to ASD (Sledziowska et al., 2020).
- Line no 175-177 “These studies prompted…” can be rewritten for clarity of message.
As suggested by the Reviewer, we re-write that sentence as “To compare the autism-like behavior in rats induced by VPA with perinatal exposure to DE-SOA, we used rat models in the present study to examine the ASD related social behaviors and related gene expressions in ASD related brain region, prefrontal cortex.” In our revised manuscript.
- Line no 151-152 “We did not know… not clear, please rewrite.
As commented by the Reviewer, we re-write that sentence as “We found no effect of DE on social novelty preference performance and social behavior-related genes such as 5HT5B, BDNF, IL-1 b, HO1 and glutamate in the present study.” in our revised manuscript.
- Line no 77-80 and 198-199, authors wrote about unpublished data. These days many journals do not allow to include unpublished data. In my view authors should remove findings of unpublished data.
  As suggested by the Reviewer, we deleted “unpublished data” in our revised manuscript.
- Recent reports are few….it is good to added/cite recent findings.
As suggested by the Reviewer, we added recent findings [Ref. 26, 27, 60 and 62] in our revised manuscript.

Reviewer 2 Report
In this study, autism-like behavior was observed in rats after exposing them to diesel exhaust or diesel-exhaust origin secondary organic aerosol from GD14-21. In addition, several autism-typical molecular changes in the brain were measured, such as down-regulation of 5-HT5B receptors and BDNF, as well as a up-regulation of some inflammation markers.
In principle, I like this study a lot. The research topic is interesting and very actual. The study protocol is nice and the results are very convincing. However, the manuscript needs some revision. First, careful language-editing is needed (for a few examples, see below). Furthermore, I think that a different apporach in the statistical analyses should be used.
More specific comments:
l.16: Consider to change "social behavior" in "autism-like behavior" or "behavioral enfophenotypes of ASD".
l.39: Consider re-editing this sentence. For example, "In animal studies, male mice (...) expressed features of ASD..."
Figure 1B: For consistency with the sketch on top of the panel, the authors should consider changing the order of the bars. First red (left in the sketch) and then green (right in the sketch).
l.114f: To be consistent with figure 2, consider using "% Win" instead of "win%".
l.141: In this sentence, the species changes from rats to mice. Please correct.
l.146: Please correct/re-edit the caption of Figure 4. Panel B is not Nlgn3 but 5HT5B, C is not BDNF but Nlgn3, D-F is not described. According to the text, the upper panels are males, and the lower ones females which shoudl be stated in the caption, too.
l.180: In don't understand the sentence "The reason for using rat model...". Why are these reasons? Of course, for example, sex differences, BDNF, etc. can also be studied in mice. For me, using rats is totally okay. However, if the authors want to state a specific rationale, this rationale should be comprehensible.
l.199: Why not stating in shich sex poor sociability was found?
Discussion: If animal studies are cited, they are very often from the author's group. Please consider citing piblications from other groups, too.
l.251: "We did not know the no effect..." Please consider re-editing. Independent of the language,I have an issue with the content. DE clearly had an effect of social novelty preference since there was no preference (which is present in the control group).
l.257: Please used "have" instead of "are". Or "are of".
l.280: Here, again "mice" appear.
l.401: In my opinion, the data should be analysed by two-way ANOIVA using sex and exposure type as factors. Also, data should be checked for normal distribution.
l.418. "...performed the behavioral tests..."
Author Response
Reviewer # 2
In this study, autism-like behavior was observed in rats after exposing them to diesel exhaust or diesel-exhaust origin secondary organic aerosol from GD14-21. In addition, several autism-typical molecular changes in the brain were measured, such as down-regulation of 5-HT5B receptors and BDNF, as well as a up-regulation of some inflammation markers.
In principle, I like this study a lot. The research topic is interesting and very actual. The study protocol is nice and the results are very convincing. However, the manuscript needs some revision. First, careful language-editing is needed (for a few examples, see below). Furthermore, I think that a different approach in the statistical analyses should be used.
More specific comments:
l.16: Considered to change "social behavior" in "autism-like behavior" or "behavioral enfophenotypes of ASD".
As suggested by the Reviewer, we changed to “behavioral phenotypes of ASD” in our revised manuscript.
l.39: Consider re-editing this sentence. For example, "In animal studies, male mice (...) expressed features of ASD..."
As commented by the Reviewer, we re-write that sentence as “In animal studies, the male mice exposed to ambient ultrafine particles during early postnatal period that is equivalent to human third trimester expressed features of ASD including learning and memory deficit, repetitive impulsive-like behavior, neuroinflammation, microglial activation, ventriculomegaly and excitatory/inhibitory imbalance [8, 9].” in our revised manuscript.
Figure 1B: For consistency with the sketch on top of the panel, the authors should consider changing the order of the bars. First red (left in the sketch) and then green (right in the sketch).
As commented by the Reviewer, we changed the order of bar in Figure 2 (B) in our revised manuscript.
l.114f: To be consistent with figure 2, consider using "% Win" instead of "win%".
As commented by the Reviewer, we corrected “Win%” to “%Win” in line 121 in our revised manuscript.
l.141: In this sentence, the species changes from rats to mice. Please correct.
We would like to thank the Reviewer for pointing out our mistake. We corrected male rats and female rats in our revised manuscript.
l.146: Please correct/re-edit the caption of Figure 4. Panel B is not Nlgn3 but 5HT5B, C is not BDNF but Nlgn3, D-F is not described. According to the text, the upper panels are males, and the lower ones females which should be stated in the caption, too.
We would like to thank the Reviewer for pointing out our mistake. We correct the caption of Figure 4 in our revised manuscript.
l.180: In don't understand the sentence "The reason for using rat model...". Why are these reasons? Of course, for example, sex differences, BDNF, etc. can also be studied in mice. For me, using rats is totally okay. However, if the authors want to state a specific rationale, this rationale should be comprehensible.
Our research group has reported social behavior using mice model in Ref. 24 (Front. Neurosci). In the present study, we used rat model to compare behavioral phenotypes of ASD in VPA-induced autism rat model and Fmr1Ko and Nlgn3 Ko rats in further study. For that reason, we explained using rat model in the present study as follows;
To compare the autism-like behavior in rats induced by VPA [25] with perinatal exposure to DE-SOA, we used rat models in the present study to examine the ASD related social behaviors and related gene expressions in ASD related brain region, prefrontal cortex. In 3 chamber social behavioral test, we found same pattern of sociability and social novelty preference were observed in male mice and rats. Our targets in previous mice study and the present rat study were different in animal strain, target brain area and target genes. The other reasons for using rat model in the present study were to examine the ASD related social behaviors other than 3 chamber test, sexual different effects and role of social behavior related genes such as serotonin, BDNF and inflammatory marker IL-1 b.
l.199: Why not stating in which sex poor sociability was found?
We deleted that unpublished data in our revised manuscript.
Discussion: If animal studies are cited, they are very often from the author's group. Please consider citing publications from other groups, too.
As suggested by the Reviewer, we added recent findings [Ref. 26, 27, 60 and 62]in our revised manuscript.
l.251: "We did not know the no effect..." Please consider re-editing. Independent of the language, I have an issue with the content. DE clearly had an effect of social novelty preference since there was no preference (which is present in the control group).
We re-write that sentence in line 257 in our revised manuscript. DE exposed both male female rats show no statistically significant effect on sociability and social novelty preference.
l.257: Please used "have" instead of "are". Or "are of".
As suggested by the Reviewer, we corrected are to have as “Recently, we have been investigating the candidate chemicals which have adverse effects in DE and DE-SOA in inhalation chambers.” in our revised manuscript.
l.280: Here, again "mice" appear.
We would like to thank the Reviewer for pointing out our mistake. We corrected “mice” to “rats” in our revised manuscript.
l.401: In my opinion, the data should be analysed by two-way ANOIVA using sex and exposure type as factors. Also, data should be checked for normal distribution.
Actually, we have analyzed our data by two-way ANOVA using sex and exposure type as factors. We observed exposure effect only and did not observe sex effect.
l.418. "...performed the behavioral tests..."
As commented by the Reviewer, we corrected “investigated” to “performed” in our revised manuscript.
